# Design, Synthesis and Biological Evaluation of New Carbohydrate-Based Coumarin Derivatives as Selective Carbonic Anhydrase IX Inhibitors via “Click” Reaction

**DOI:** 10.3390/molecules27175464

**Published:** 2022-08-25

**Authors:** Naying Chu, Yitong Wang, Hao Jia, Jie Han, Xiaoyi Wang, Zhuang Hou

**Affiliations:** 1Department of Pharmacy, The First People’s Hospital of Shangqiu, Suiyang District, 292 Kaixuan Road, Shangqiu 476000, China; 2China-Japan Research Institute of Medical and Pharmaceutical Sciences, Shenyang Pharmaceutical University, 103 Wenhua Road, Shenyang 110016, China; 3Key Laboratory of Structure-Based Drugs Design and Discovery (Ministry of Education), Shenyang Pharmaceutical University, Shenyang 110016, China

**Keywords:** carbonic anhydrase IX inhibitors, coumarin, one-pot click chemistry, carbohydrate

## Abstract

In this work, we designed a series of new carbohydrate-based coumarin carbonic anhydrase IX inhibitors by using 1,2,3-triazoles as linker. Next, these designed compounds were synthesized by the optimized one-pot click chemistry reaction condition. Subsequently, these target compounds were assayed for the inhibition of three carbonic anhydrase isoforms (CA I, CA II and CA IX). Intriguingly, all the compounds showed better CA IX inhibitory activity than initial coumarin fragments. Among them, compound **10a** (IC_50_: 11 nM) possessed the most potent CA IX inhibitory activity, which was more potent than the reference drug acetazolamide (IC_50_: 30 nM). Notably, compound **10a** showed 3018-fold, 1955-fold selectivity relative to CA I and CA II, respectively. Meanwhile, representative compounds could reduce tumor cell viability and the extracellular acidification in HT-29 and MDA-MB-231 cancer cell lines. Even more interestingly, our target compounds had no apparent cytotoxicity toward MCF-10A cell line. In addition, the in vitro stability assays also indicated our developed compounds possessed good liver microsomal metabolic stabilities and plasma stability. Furthermore, representative compounds revealed relatively low hERG cardiac toxicity and acute toxicity. Furthermore, docking studies were carried out to understand the interactions of our target compounds with the protein target CA IX. Collectively, our results suggest that compound **10a**, as a selective CA IX inhibitor, could be an important lead compound for further optimization and development as an anticancer agent.

## 1. Introduction

Carbonic anhydrases (CAs, EC 4.2.1.1) are a very omnipresent zinc metalloenzymes, which are encoded by eight families, namely α-, β-, γ-, δ-, ζ-, η-, θ and ι. The α-CAs are represented by 15 different isoforms of human carbonic anhydrase (hCA) varies from CA I-CA XIV [1,2,3,4,5,6]. The expression of CAs is widely distributed in various tissues and they play an important role in numerous physiological and pathological processes, such as pH and CO_2_ homeostasis, respiration and transport of CO_2_ and HCO_3^−^_, biosynthetic reactions, tumorigenicity, bone resorption, calcification, etc., [7,8,9,10]. The transmembrane isoform CA IX has been characterized as biomarkers for several tumors. CA IX is not expressed in healthy tissues, but it is overexpressed in tumor microenvironment. CA IX is also one of the best markers for cellular hypoxia, and it promotes the acidification of the extracellular matrix, which is conducive to the growth and metastasis of tumor cells [11,12,13,14]. Therefore, CA IX is considered as a potential antitumor drug target.

The classical carbonic anhydrase inhibitors (CAIs) are sulfonamides and their derivatives. These sulfonamide CAIs have been used clinically to treat glaucoma, edema, epilepsy, and altitude sickness for a long time, but their key barriers relate to the 15 isoforms in humans, their diffuse localization in many tissues and organs, the lack of isozyme selectivity. The available CAIs act systemically and bind nonspecifically, causing a range of undesired side effects, due to off-target inhibition [15,16]. Thus, selective non-classical CAIs are urgently needed. In recent years, coumarin derivatives were reported as a promising new class CAIs which do not directly interact with the zinc ion from the CA active site, and the mechanism of action has been clarified [17,18,19,20,21,22,23,24,25]. Due to the selective inhibition on hCA IX, as shown in Figure 1, coumarins and their isosteres represent very interesting classes of CAIs.

Among them, the natural product coumarin **1** acted as an effective CAI against CA I and CA II. In addition, it showed medium inhibitory activity against CA IX. The simple nonsubstituted derivative **2** is generally very poor CA IX inhibitor. As CAIs isosters of the simple coumarin **2**, thiocoumarin, 2-thioxocoumarin, and dithiocoumarin were reported and these isosters showed improved inhibitory activity against CA IX. It is worth noting that these structures showed only micromolar level of inhibitory activity against CA IX. Interestingly, 7-hydroxy coumarin (umbelliferone) and 4-methyl-7-hydroxy coumarin (4-methylumbelliferone) not only showed significantly selective for CA IX, but also possessed more potent inhibitory activity against CA IX [26,27,28]. Therefore, structural modification of these simple coumarins need to be performed to improve the inhibitory effect on hCA IX. For another example, glycosyl coumarins **8** and **9** showed low nanomolar inhibitory activity against CA IX [29].

It is widely known that coumarins as hydrophobic fragment could interact with the hydrophobic pocket at the entrance of the CA active site cavity. However, the neighboring hydrophilic subpocket cannot be fully occupied by these simple coumarins. Therefore, the CA IX inhibitory activity may be improved when some lipophilic bulky moieties was introduced into coumarins. The tail approaches have been extensively utilized in the design of CAIs, especially sugar tail approaches [30,31,32]. In our previous study, the introduction of the sugar moiety led to a significant enhancement of the CA inhibitory activity. As an extension of coumarins-based CAIs, we designed and synthesized a series of glycosyl conjugates of coumarins which were assessed for their inhibitory actions against three hCAs such as the hCA I, II, IX. Next, the inhibition of proliferation, the extracellular acidification of tumor cells was explored. Meanwhile, the metabolic stability of target compounds was evaluated. Finally, the hERG cardiac toxicity and acute toxicity were also investigated.

## 2. Results

### 2.1. Design of New Carbohydrate-Based Coumarin Derivatives

The target compounds comprised three main elements: (i) a sugar-tail scaffold, (ii) a coumarin moiety, and (iii) a 1,2,3-triazole linker. Due to possessing the CA IX selectivity, the coumarin moieties was chosen as initial fragments. In order to form more hydrogen bonds with the enzymatic cavity, different monosaccharide scaffolds were chosen as the hydrophilic groups for drawing favorable interaction with the hydrophilic halves of the CA active site. The 1,2,3-triazole ring is a bioisostere of amide group, endowed with a moderate dipole character, hydrogen bonding capability, rigidity, stability in the in vivo environment, and shows good tolerance to metabolic processes as well as to pH fluctuations [33,34,35,36,37]. Therefore, the 1,2,3-triazole played the role as a biocompatible covalent linker between the sugar and coumarin moiety. The approach is shown in Figure 2. 

### 2.2. Chemistry

In order to prepare the designed compounds, we decided to develop an one-pot synthesis of 1,2,3-triazoles from benzoyl glycosyl bromides such as **13a**, sodium azide, and coumarin-derived alkynes. Among the most common monosaccharides, D-glucose **11a** was utilized as the starting material to synthesize model substrate **13a**. As described in Figure 1, the intermediate **12a** was produced via benzoylation of **11a**. The intermediate **12a** was treated with HBr-AcOH to obtain the intermediate **13a**. In addition, coumarin-derived alkynes **14a**–**14b** were efficiently synthesized in high yields (Figure 2).

The model reaction, as shown in Table 1, was performed in the presence of different types of solvents, copper sulfate at different reaction temperature and time to find optimum conditions. Notably, this reaction did not provide the desired product **15a** when *t*-BuOH, DMSO, DMF were chosen as solvent (Table 1, entries 1–3). When H_2_O was used as solvent, the desired product was observed (Table 1, entry 4). These preliminary results encouraged us to do further reaction conditions screening to increase the yield of **15a**. Subsequently, increasing the reaction temperature and prolonging the reaction time could not improve the yield (Table 1, entry 5–9). It is worth noting that the hydrolysate of compound **13a** was found due to the decomposition of benzoyl glycosyl bromide by water. Therefore, H_2_O is not suitable as a solvent alone. Next, the use of mixed solvents can significantly increase the yield such as H_2_O/*t*-BuOH, H_2_O/DMF and H_2_O/DMSO (Table 1, entry 10–12). Among these three mixed solvents, H_2_O/*t*-BuOH provides the highest yield (42%). However, at present, the obtained yields are not satisfactory to us due to the partial hydrolysis of benzoyl glycosyl bromide **13a**. Interestingly, the yield can be further improved by adjusting the ratio of *t*-BuOH and H_2_O (Table 1, entry 13–15). The best results were obtained when using H_2_O/*t*-BuOH (1:3). Several other catalysts, including CuBr and CuSO_4_·5H_2_O/VcNa were next evaluated and CuSO_4_·5H_2_O/VcNa was found to have higher catalytic activity (Table 1, entry 16–17). Subsequently, the yield of the product **15a** was increased to 90% when the reaction temperature was increased to 70 °C (Table 1, entry 18–20). In addition, the reaction still has a high yield even if the reaction time is reduced to 30 min. Therefore, the optimal reaction condition entailed the use of CuSO_4_·5H_2_O/VcNa as a catalyst in H_2_O/*t*-BuOH (1:3) at 70 °C.

To study the scope of the above reaction, a wide range of diversely benzoyl glycosyl bromides were tried. To our delight, under the conditions mentioned above, two coumarin-derived alkynes could react smoothly with the in situ generated organic azides from benzoyl glycosyl bromides and NaN_3_ to give the corresponding products in excellent yields (Table 2).

The target compound **10a** was obtained by deprotection of benzoyl groups using NaOMe/MeOH in high yield (Figure 3) [38]. All other target compounds **10b**–**10p** were also prepared by this procedure. Since there are two possible configurations (α and β) of the C1-hydrogen, the configuration of representative compound **10a** was studied by the data of ^1^H-NMR(see Appendix A) in this paper. The data of ^1^H-NMR showed that the chemical shift of C1-hydrogen (double peak) appears in 5.57 ppm, the coupling constant is 9.3 Hz. These results are consistent with the β configuration.

### 2.3. Biological Activity

#### 2.3.1. Carbonic Anhydrase Inhibition

The CA inhibitory activities of compounds **10a**–**10p** were measured against three isoforms hCA I, II, and IX by the esterase assay. Meanwhile, acetazolamide (AZA), umbelliferone and 4-methylumbelliferone served as standard inhibitors were tested. The CA inhibition data are shown in Table 3. The selectivity ratios for CA IX inhibition over CA I and CA II are also presented in Table 3.

(i) All new compounds exhibited weak inhibitory activity against the cytosolic isoform hCA I, with IC_50_ in the micromolar range (from 17.5 to 65.6 μM). In addition, this isoform was not at all inhibited by the parent coumarins umbelliferone and 4-methylumbelliferone.

(ii) The physiologically dominant isoform hCA II was similarly poorly inhibited by all the compounds, which showed IC_50_ values in the range of 12.2–32.8 μM. The unsubstituted coumarin showed no inhibitory power against hCA II again. As CA I and CA II are considered important off-target CA isozyme, the poor inhibition against CA I and CA II is an interesting feature for compounds designed to target the tumor-associated enzymes.

(iii) The tumor-associated isoform hCA IX was weakly inhibited by parent coumarins umbelliferone and 4-methylumbelliferone. Notably, the introduction of sugar moieties into the coumarin ring at position 7 lead to a significant increase of the hCA IX inhibitory potency with IC_50_ in the range of 0.011–0.132 μM. Among them, the glucose derivative **10a** and mannose derivative **10c** showed most potent hCA IX inhibitory activities with IC_50_ values of 11 nM in the case of **10a** and 15 nM in the case of **10c**. The increase in inhibitory potency against this isoform may be due to the matching the hydrophilic half of the active site with the hydrophilic glucosyl moiety, and in addition to interacting with hydrophobic half of the active site with coumarins.

(iv) Next, the isoform selectivities of compounds **10a**–**10p** against CAI, CAII and CAIX were investigated. The results indicated that the target compounds showed good CA IX isoform selectivity (up to 3018-fold, 1955-fold selectivity toward the hCA I and CA II, respectively). The selective inhibition of carbonic anhydrase IX of coumarin-derivatives might decrease the side effects, and these target compounds were expected to be interesting lead compounds for treatment of cancer.

#### 2.3.2. Effect of the Synthetic Compounds on the Cell Viability of Human Cancer Cell Lines

In the present study, the potential cytotoxicity of compounds **10a**–**10p** was evaluated in vitro by MTT assay. As a matter of fact, HT-29 cells possessed a high CA IX expression under ambient air, while MDA-MB-231 cells showed enhanced CA IX expression only upon hypoxia. So, half of each cell line were subjected to hypoxic conditions (0.5% O_2_, 5% CO_2_ and 94.5% N_2_) for inducing the expression of CA IX and the other half were incubated under normal conditions (normoxia). Subsequently, all synthesized derivatives were screened at a concentration of 100 μM to test their cytotoxicity in mentioned above cells.

Results in Table 4 indicated all the compounds were capable of reducing tumor cell viability in these two cancer cell lines under both of the conditions. At normal oxygen content, the inhibitory activity of compounds against HT-29 cancer cells was slightly higher than that in MDA-MB-231 cells. Affected by hypoxic conditions which leading to overexpression of CA IX, the inhibitory activity of compounds against the same tumor cells was higher than that under normal oxygen. The inhibitory activity of compounds against HT-29 under hypoxia conditions was similar to that of MDA-MB-231 cells. What’s more, note that new carbohydrate-based coumarins designed in this study exhibited comparatively potent inhibitions at the enzyme level. Whereas weak cellular antitumor activities (high μM) of these compounds reported in this study were also observed, which revealed that hCA IX inhibitors are not potent enough in anti-tumor monotherapy only depending on the regulation of tumor microenvironment. Therefore, more pharmacological effects of CA IX inhibitors as antitumor agents need to be further explored. Since CA IX can regulate the extracellular pH of tumor cells, we next examined whether our inhibitors had any effect on the extracellular pH of tumor cells.

#### 2.3.3. Extracellular pH Measurement of Human Cancer Cell Lines in the Presence of Compounds **10a** and **10h**

One important feature of CA IX is regulating the pH of tumor cells. The expression of CA IX in solid tumors will make the extracellular pH become more acidic to promote tumor growth and metastasis. Therefore, the effect on extracellular pH of the representative compounds **10a** and **10h** was tested in hypoxia and normoxic conditions. The results showed that most potent compound **10a** had an important impact on the extracellular acidification of cancer cells in two different concentrations (0.1 and 0.5 mM) (Figure 3). Especially, the extracellular acidification of these two cancer cells could be slightly reversed by this compound under normoxic condition. In addition, the reversal of extracellular acidification is more significant under hypoxia condition. Furthermore, a dose-dependent effect was observed. In addition, compared with the most potent compound **10a**, the compound **10h** with weak enzymatic activity showed weaker reversal effect of extracellular acidification on both cell lines in hypoxia and normoxic conditions.

#### 2.3.4. In Vitro Cytotoxicity Studies on Normal Mammalian Cells

To obtain the preliminary safety profiles of these carbohydrate-based coumarins in normal human cells, compounds **10a** and **10c** were chosen to evaluated for cytotoxicity against human normal cell line MCF-10A by measuring its IC_50_ in MTT assay. The data (Table 5.) suggested the carbohydrate-based coumarins **10a** and **10c** had no apparent cytotoxicity toward MCF-10A cell line (IC_50_ >100 μM).

#### 2.3.5. In Vitro Liver Microsomal Stability Assay

A perfect drug should be stable enough to reach its pharmacological target. Therefore, target compounds were subjected to evaluate the in vitro metabolic stability. As presented in Table 6, all target compounds exhibited excellent metabolic stability with half-life values range from 1153 min to 1843 min. Therefore, target compounds are worthy of further evaluation.

#### 2.3.6. Plasma Stability of Compound **10a**

In fact, most drugs reach their targets to show their pharmacological activity via the circulation system, therefore, the plasma stability of the target compound is a crucial consideration for the evaluation of its druggability. Therefore, the in vitro plasma stability assay of the representative compound was also performed in this study. The results revealed that compound **10a** possessed high plasma stability, with a half-life value greater than 1230 min (Table 7).

#### 2.3.7. Inhibition Evaluation on hERG Channel

The hERG channel is a potassium ion channel that plays an important role in the regulation of cardiac repolarization. Blockage of the hERG channel can cause serious cardiac side effects, such as prolongation of the QT interval. For this reason, the most potent compounds **10a** and **10c** was further assessed for inhibition of the hERG channel. As shown in Table 8, compounds **10a** and **10c** showed weak inhibition (IC_50_ > 50 μM). The results indicated that compounds **10a** and **10c** were less likely to produce cardiotoxicity.

#### 2.3.8. Preliminary Assessment of the Acute Toxicity

Finally, acute toxicity of representative compounds was conducted in mice to preliminarily evaluate the risk of target compounds to mammals. The compounds **10a** and **10c** were given by intragastric administration from 500 mg/kg to 1500 mg/kg, respectively. Animals were observed for lethality for 14 days. The obtained preliminary data (Table 9) allowed us to conclude that LD_50_ values for these compounds were expected to exceed 1500 mg/kg, because after administering this dose none of the animals died.

#### 2.3.9. Molecular Docking Analyses

In order to predict the interact modes between target compounds and hCA IX, molecular docking studies were employed in this study. At first, a redocking experiment for AAZ with CA IX (PDB:3IAI) was processed by using AutoDock 4.2.6 (Developed and maintained by the Olson Laboratory at the Scripps Research Institute, La Jolla, CA, USA) with the AutoDock4Zn force field. The redocking results showed that the crystal pose of AAZ was reproduced fairly well. Next, the same models were applied to the previous reported sugar-containing coumarins **8** and **9**. As seen from Figure 4, the hydrophilic and hydrophobic region of active site were displayed with blue and red background, respectively. The obtained results indicated that the coumarin groups of the compounds **8** and **9** could bind to the hydrophobic part of the active site while the sugar fragments failed to penetrate deep into the hydrophilic part to form sufficient binding because they were far away from the hydrophilic part. The most potent compounds **10a** and **10c** occupied the hydrophilic and hydrophobic region of active site simultaneously. More specifically, the sugar groups and 1,2,3-trizaole group of the compounds could bind to the hydrophilic part while the coumarin groups could bind to the hydrophobic part.

## 3. Materials and Methods

### 3.1. Chemistry

Unless otherwise noted, all solvents and reagents were obtained from commercial suppliers and used without further purification. ^1^H-NMR and ^13^C-NMR spectra were measured with an AV-600 spectrometer (Bruker Bioscience, Billerica, MA, USA), with tetramethyl silane as an internal standard. The 200–300 mesh silica gel (Qingdao Haiyang Chemical, Qingdao, China) was used for Column chromatography. TLC analysis was performed on silica gel plates GF254 (Qingdao Haiyang Chemical, Qingdao, China). ESI-MS were obtained on Agilent ESI-QTOF instrument (Agilent, Santa Clara, CA, USA). High resolution mass spectra (HRMS) were recorded on an Agilent Accurate-Mass Q-TOF 6530 (Agilent, Santa Clara, CA, USA) in ESI mode.

#### General Procedure for the Synthesis of Target Compound **10a**

To a solution of **11a** (5 g, 27.8 mmol) in pyridine (50 mL) was added benzoyl chloride (16.1 mL, 139 mmol) under the ice bath and stirred for 5 h at room temperature. The H_2_O (100 mL) and dichloromethane (DCM, 100 mL) were added and the mixture was stirred for 20 min. Next, the mixture was extracted by CH_2_Cl_2_ and the organic layer was washed with 0.1 M HCl (20 mL), saturated NaHCO_3_ (30 mL). In addition, the organic layer was washed with water, then dried over anhydrous Na_2_SO_4_ and concentrated under reduced pressure to give **12a**. To a solution of **12a** in dichloromethane (50 mL) was added HBr-HOAc (35 mL). The reaction mixture was stirred for 5 h. Next, H_2_O (50 mL) was added and the reaction mixture was vigorously stirred for 20 min. The organic layer was washed with saturated NaHCO_3_ (30 mL), and then dried over anhydrous Na_2_SO_4_ and concentrated under reduced pressure. The crude products were purified by silica gel column chromatography eluting with 20% *v*/*v* EtOAc/n-hexane to give **13a** as a white solid (16.8 g, 92%). To a solution of **13a** (0.66 g, 1 mmol) in H_2_O/t-BuOH (1:3) (20 mL) was added NaN_3_ (0.13 g, 2 mmol), 7-(Prop-2-ynyloxy)-*2H*-chromen-2-one (0.2 g, 1 mmol), CuSO_4_·5H_2_O (25.0 mg, 0.1 mmol) and sodium ascorbate (59.4 mg, 0.3 mmol). The reaction mixture was vigorously stirred at 70 °C for 30 min. The solvent t-BuOH was removed under vacuo and the rest of the reaction mixture was extracted by using ethyl acetate. Furthermore, the organic layer was washed with water, then dried over anhydrous Na_2_SO_4_ and concentrated under reduced pressure. The crude products were purified by silica gel column chromatography eluting with 25% *v*/*v* acetone/n-hexane to obtain **15a** (0.76 g, 0.93 mmol). Next, the benzoyl group was removed by sodium methoxide to acquire a crude residue which was purified by column chromatography eluting with 10% *v*/*v* MeOH/CH_2_Cl_2_ to obtain target compound **10a** (0.37 g, 99%) as a white solid. The compounds **10b**–**10p** were synthesized by the above synthetic method.

*7-[(1-β-. D**-glucopyranosyl-1H-1,2,3-triazol-4-yl) methoxy]-2H-chromen-2-one* (**10a**). m.p. 122.5–124.1 °C; ^1^H NMR (600 MHz, DMSO-*d*_6_) δ 8.50 (s, 1H), 7.99 (d, *J* = 9.5 Hz, 1H), 7.65 (d, *J* = 8.6 Hz, 1H), 7.18 (d, *J* = 2.1 Hz, 1H), 7.04 (dd, *J* = 8.6, 2.3 Hz, 1H), 6.30 (d, *J* = 9.5 Hz, 1H), 5.57 (d, *J* = 9.3 Hz, 1H), 5.42 (d, *J* = 6.0 Hz, 1H), 5.29 (d, *J* = 5.0 Hz, 1H), 5.27 (s, 2H), 5.16 (d, *J* = 5.5 Hz, 1H), 4.63 (t, *J* = 5.5 Hz, 1H), 3.78 (td, *J* = 9.1, 6.2 Hz, 1H), 3.72–3.68 (m, 1H), 3.45 (dd, *J* = 14.3, 5.6 Hz, 2H), 3.39 (td, *J* = 8.9, 5.1 Hz, 1H), 3.24 (dt, *J* = 14.5, 7.3 Hz, 1H). ^13^C NMR (151 MHz, DMSO-*d*_6_) δ 161.61, 160.75, 155.80, 144.78, 142.35, 130.03, 124.76, 113.32, 113.18, 113.10, 102.00, 87.99, 80.45, 77.41, 72.54, 70.04, 62.05, 61.21. ESI-MS (*m*/*z*): 428.1 [M + Na] ^+^; HRMS (ESI): Calcd for [M + Na] ^+^ C_18_H_19_N_3_NaO_8_:428.1077, Found 428.1097.

*7-[(1-β-. D-galactopyranosyl-1H-1,2,3-triazol-4-yl) methoxy]-2H-chromen-2-one* (**10b**). m.p. 128.6–129.9 °C; ^1^H NMR (600 MHz, DMSO-*d*_6_) δ 8.43 (s, 1H), 8.00 (d, *J* = 9.5 Hz, 1H), 7.65 (d, *J* = 8.6 Hz, 1H), 7.19 (d, *J* = 2.3 Hz, 1H), 7.04 (dd, *J* = 8.6, 2.4 Hz, 1H), 6.30 (d, *J* = 9.5 Hz, 1H), 5.52 (d, *J* = 9.2 Hz, 1H), 5.28 (s, 2H), 5.27 (d, *J* = 6.2 Hz, 1H), 5.04 (d, *J* = 5.3 Hz, 1H), 4.70 (t, *J* = 5.6 Hz, 1H), 4.65 (d, *J* = 5.3 Hz, 1H), 4.05 (td, *J* = 9.2, 5.9 Hz, 1H), 3.78–3.75 (m, 1H), 3.72 (t, *J* = 6.1 Hz, 1H), 3.57–3.54 (m, 1H), 3.53–3.51 (m, 1H), 3.50–3.48 (m, 1H). ^13^C NMR (151 MHz, DMSO-*d*_6_) δ 161.63, 160.74, 155.81, 144.78, 142.44, 130.03, 124.39, 113.32, 113.17, 113.09, 102.01, 88.61, 78.93, 74.15, 69.79, 68.94, 62.06, 60.91. ESI-MS (*m*/*z*): 428.1 [M + Na] ^+^; HRMS (ESI): Calcd for [M + Na] ^+^ C_18_H_19_N_3_NaO_8_: 428.1090, Found 428.1070.

*7-[(1-β-. D-mannopyranosyl-1H-1,2,3-triazol-4-yl) methoxy]-2H-chromen-2-one* (**10c**). m.p. 121.5–122.6 °C; ^1^H NMR (600 MHz, DMSO-*d*_6_) δ 8.39 (d, *J* = 5.0 Hz, 1H), 8.00 (dd, *J* = 9.4, 5.0 Hz, 1H), 7.68–7.62 (m, 1H), 7.21–7.16 (m, 1H), 7.07–7.02 (m, 1H), 6.34–6.28 (m, 1H), 6.05 (d, *J* = 3.9 Hz, 1H), 5.34–5.31 (m, 1H), 5.29 (d, *J* = 4.5 Hz, 2H), 5.06 (s, 1H), 5.00 (s, 1H), 4.62 (d, *J* = 4.7 Hz, 1H), 3.90 (d, *J* = 2.8 Hz, 1H), 3.78–3.72 (m, 1H), 3.62 (s, 1H), 3.49 (dd, *J* = 10.7, 6.7 Hz, 2H), 3.42 (d, *J* = 4.1 Hz, 1H). ^13^C NMR (151 MHz, DMSO-*d*_6_) δ 161.66, 160.75, 155.80, 144.77, 141.91, 130.01, 113.33, 113.16, 113.07, 102.01, 86.46, 80.84, 73.58, 70.90, 66.69, 62.07, 61.55. ESI-MS (*m*/*z*): 428.1 [M + Na] ^+^; HRMS (ESI): Calcd for [M + Na] ^+^ C_18_H_19_N_3_NaO_8_:428.1070, Found 428.1077.

*7-[(1-β-. D-glucosaminogly-1H-1,2,3-triazol-4-yl) methoxy]-2H-chromen-2-one* (**10d**). m.p. 131.3–134.2 °C; ^1^H NMR (600 MHz, DMSO-*d*_6_) δ 8.50 (s, 1H), 8.03 (d, *J* = 8.8 Hz, 1H), 7.71 (d, *J* = 8.8 Hz, 1H), 7.17 (d, *J* = 2.3 Hz, 1H), 7.06 (dd, *J* = 8.8, 2.4 Hz, 1H), 6.23 (s, 1H), 5.51 (d, *J* = 9.5 Hz, 1H), 5.45 (s, 1H), 5.31–5.25 (m, 3H), 4.67 (s, 1H), 3.68 (dd, *J* = 11.4, 3.5 Hz, 1H), 3.46 (dd, *J* = 11.6, 5.3 Hz, 1H), 3.44–3.41 (m, 1H), 3.27 (dt, *J* = 28.2, 8.9 Hz, 2H), 3.13 (t, *J* = 9.3 Hz, 1H), 1.52 (s, 2H). ^13^C NMR (151 MHz, DMSO-*d*_6_) δ 162.03, 161.11, 155.66, 154.41, 142.97, 127.54, 125.28, 114.38, 113.54, 112.29, 102.56, 89.52, 81.05, 77.97, 70.58, 62.59, 61.70, 57.18. ESI-MS (*m*/*z*): 427.1 [M + Na] ^+^; HRMS (ESI): Calcd for [M + Na] ^+^ C_18_H_20_N_4_NaO_7_: 427.1332, Found 427.1397.

*7-[(1-β-. D-xylopyranosyl-1H-1,2,3-triazol-4-yl) methoxy]-2H-chromen-2-one* (**10e**). m.p. 128.4–130.6 °C; ^1^H NMR (600 MHz, DMSO-*d*_6_) δ 8.49 (s, 1H), 8.00 (d, *J* = 9.5 Hz, 1H), 7.65 (d, *J* = 8.6 Hz, 1H), 7.18 (d, *J* = 2.4 Hz, 1H), 7.04 (dd, *J* = 8.6, 2.4 Hz, 1H), 6.31 (d, *J* = 9.5 Hz, 1H), 5.52 (d, *J* = 9.3 Hz, 1H), 5.44 (d, *J* = 6.1 Hz, 1H), 5.32 (d, *J* = 4.9 Hz, 1H), 5.28 (s, 2H), 5.18 (d, *J* = 5.0 Hz, 1H), 3.84 (dd, *J* = 11.1, 5.3 Hz, 1H), 3.78 (td, *J* = 9.1, 6.1 Hz, 1H), 3.48 (ddd, *J* = 14.2, 10.4, 5.2 Hz, 1H), 3.38 (d, *J* = 10.9 Hz, 1H), 3.37–3.35 (m, 1H). ^13^C NMR (151 MHz, DMSO-*d*_6_) δ 159.44, 158.60, 153.65, 142.63, 140.23, 127.87, 122.48, 111.21, 111.03, 110.95, 99.87, 86.48, 75.37, 70.32, 67.43, 66.68, 59.91. ESI-MS (*m*/*z*): 398.1 [M + Na] ^+^; HRMS (ESI): Calcd for [M + Na] ^+^ C_17_H_17_N_3_NaO_7_: 398.0964, Found 398.0978.

*7-[(1-β-. L-arabinopyranosyl-1H-1,2,3-triazol-4-yl) methoxy]-2H-chromen-2-one* (**10f**). m.p. 133.1–135.1 °C; ^1^H NMR (600 MHz, DMSO-*d*_6_) δ 8.42 (s, 1H), 8.00 (d, *J* = 9.5 Hz, 1H), 7.65 (d, *J* = 8.6 Hz, 1H), 7.19 (d, *J* = 2.4 Hz, 1H), 7.04 (dd, *J* = 8.6, 2.4 Hz, 1H), 6.31 (d, *J* = 9.5 Hz, 1H), 5.45 (d, *J* = 9.1 Hz, 1H), 5.30 (d, *J* = 12.2 Hz, 3H), 5.03 (s, 1H), 4.81 (s, 1H), 4.05 (td, *J* = 9.1, 4.2 Hz, 1H), 3.81 (dd, *J* = 12.4, 1.9 Hz, 1H), 3.77 (d, *J* = 11.4 Hz, 2H), 3.56 (dd, *J* = 9.2, 2.7 Hz, 1H). ^13^C NMR (151 MHz, DMSO-*d*_6_) δ 161.61, 160.74, 155.80, 144.78, 142.41, 130.02, 124.33, 113.36, 113.17, 113.08, 102.03, 88.88, 73.70, 69.81, 69.78, 68.82, 62.05. ESI-MS (*m*/*z*): 398.1 [M + Na] ^+^; HRMS (ESI): Calcd for [M + Na] ^+^ C_17_H_17_N_3_NaO_7_: 398.0964, Found 398.0976.

*7-[(1-β-. L-rhamnosyl-1H-1,2,3-triazol-4-yl) methoxy]-2H-chromen-2-one* (**10g**). m.p. 126.1–129.6 °C; ^1^H NMR (600 MHz, DMSO-*d*_6_) δ 8.34 (d, *J* = 4.3 Hz, 1H), 8.00 (dd, *J* = 9.4, 4.2 Hz, 1H), 7.65 (dd, *J* = 8.6, 4.5 Hz, 1H), 7.18 (s, 1H), 7.04 (dd, *J* = 6.6, 1.9 Hz, 1H), 6.34–6.27 (m, 1H), 6.03 (s, 1H), 5.35 (d, *J* = 4.7 Hz, 1H), 5.28 (s, 2H), 5.06–5.02 (m, 2H), 3.89 (s, 1H), 3.58 (dd, *J* = 8.4, 3.7 Hz, 1H), 3.48 (dq, *J* = 6.1, 4.9 Hz, 1H), 3.33–3.29 (m, 1H), 1.23 (d, *J* = 4.7 Hz, 3H). ^13^C NMR (151 MHz, DMSO-*d*_6_) δ 161.64, 160.75, 155.80, 144.77, 141.98, 129.99, 125.01, 113.35, 113.14, 113.05, 102.01, 86.41, 75.46, 73.36, 71.67, 71.02, 61.98, 18.31. ESI-MS (*m*/*z*): 412.1 [M + Na] ^+^; HRMS (ESI): Calcd for [M + Na] ^+^ C_18_H_19_N_3_NaO_7_: 412.1118, Found 412.1138.

*7-[(1-β-. D-ribofuranosyl-1H-1,2,3-triazol-4-yl) methoxy]-2H-chromen-2-one* (**10h**). m.p. 131.2–133.6 °C; ^1^H NMR (600 MHz, DMSO-*d*_6_) δ 8.47 (s, 1H), 8.00 (d, *J* = 9.5 Hz, 1H), 7.65 (d, *J* = 8.7 Hz, 1H), 7.17 (d, *J* = 2.3 Hz, 1H), 7.03 (dd, *J* = 8.6, 2.4 Hz, 1H), 6.31 (d, *J* = 9.5 Hz, 1H), 5.66 (d, *J* = 8.8 Hz, 1H), 5.28 (s, 2H), 5.18 (d, *J* = 3.7 Hz, 1H), 5.16 (d, *J* = 7.2 Hz, 1H), 4.92 (d, *J* = 6.1 Hz, 1H), 4.04–4.00 (m, 2H), 3.73 (dd, *J* = 6.2, 4.1 Hz, 1H), 3.68 (d, *J* = 10.5 Hz, 1H), 3.61 (dd, *J* = 9.5, 4.4 Hz, 1H). ^13^C NMR (151 MHz, DMSO-*d*_6_) δ 161.59, 160.73, 155.78, 144.77, 142.33, 130.01, 124.78, 113.35, 113.18, 113.09, 102.02, 85.39, 71.54, 69.65, 66.89, 65.65, 62.10. ESI-MS (*m*/*z*): 398.1 [M + Na] ^+^; HRMS (ESI): Calcd for [M + Na] ^+^ C_17_H_17_N_3_NaO_7_: 398.0964, Found 398.0989.

*7-[(1-β-. D-glucopyranosyl-1H-1,2,3-triazol-4-yl) methoxy]-4-methyl-2H-chromen-2-one* (**10i**). m.p. 127.9–130.2 °C; ^1^H NMR (600 MHz, DMSO-*d*_6_) δ 8.50 (s, 1H), 7.70 (d, *J* = 8.8 Hz, 1H), 7.17 (d, *J* = 2.5 Hz, 1H), 7.06 (dd, *J* = 8.8, 2.5 Hz, 1H), 6.23 (d, *J* = 1.1 Hz, 1H), 5.58 (d, *J* = 9.3 Hz, 1H), 5.42 (d, *J* = 6.0 Hz, 1H), 5.30 (d, *J* = 5.0 Hz, 1H), 5.29 (s, 2H), 5.17 (d, *J* = 5.5 Hz, 1H), 4.64 (t, *J* = 5.7 Hz, 1H), 3.79 (td, *J* = 9.1, 6.1 Hz, 1H), 3.73–3.68 (m, 1H), 3.48–3.44 (m, 2H), 3.40 (td, *J* = 8.9, 5.0 Hz, 1H), 3.24 (td, *J* = 9.1, 5.6 Hz, 1H), 2.40 (d, *J* = 0.9 Hz, 3H). ^13^C NMR (151 MHz, DMSO-*d*_6_) δ 161.51, 160.61, 155.15, 153.88, 142.39, 127.01, 124.73, 113.87, 113.01, 111.78, 102.03, 87.99, 80.45, 77.42, 72.55, 70.04, 62.04, 61.22, 18.61. ESI-MS (*m*/*z*): 442.1 [M + Na] ^+^; HRMS (ESI): Calcd for [M + Na] ^+^ C_19_H_21_N_3_NaO_8_:442.1217, Found 442.1221.

*7-[(1-β-. D-galactopyranosyl-1H-1,2,3-triazol-4-yl) methoxy]-4-methyl-2H-chromen-2-one* (**10j**). m.p. 124.2–126.6 °C; ^1^H NMR (600 MHz, DMSO-*d*_6_) δ 8.43 (s, 1H), 7.70 (d, *J* = 8.8 Hz, 1H), 7.18 (d, *J* = 2.5 Hz, 1H), 7.07 (dd, *J* = 8.8, 2.5 Hz, 1H), 6.23 (d, *J* = 1.1 Hz, 1H), 5.52 (d, *J* = 9.2 Hz, 1H), 5.29 (s, 2H), 5.27 (d, *J* = 6.0 Hz, 1H), 5.04 (d, *J* = 5.7 Hz, 1H), 4.70 (t, *J* = 5.7 Hz, 1H), 4.65 (d, *J* = 5.4 Hz, 1H), 4.05 (td, *J* = 9.3, 6.0 Hz, 1H), 3.78–3.76 (m, 1H), 3.73 (t, *J* = 6.1 Hz, 1H), 3.58–3.51 (m, 2H), 3.51–3.47 (m, 1H), 2.40 (d, *J* = 1.0 Hz, 3H). ^13^C NMR (151 MHz, DMSO-*d*_6_) δ 161.53, 160.61, 155.15, 153.89, 142.48, 127.02, 124.36, 113.86, 113.01, 111.78, 102.04, 88.61, 78.93, 74.15, 69.79, 68.94, 62.03, 60.91, 18.62. ESI-MS (*m*/*z*): 442.1 [M + Na] ^+^; HRMS (ESI): Calcd for [M + Na] ^+^ C_19_H_21_N_3_NaO_8_: 442.1226, Found 442.1254.

*7-[(1-β-. D-mannopyranosyl-1H-1,2,3-triazol-4-yl) methoxy]-4-methyl-2H-chromen-2-one* (**10k**). m.p. 121.5–122.6 °C; ^1^H NMR (600 MHz, DMSO-*d*_6_) δ 8.50 (s, 1H), 7.69 (d, *J* = 8.8 Hz, 1H), 7.16 (d, *J* = 2.5 Hz, 1H), 7.05 (dd, *J* = 8.8, 2.5 Hz, 1H), 6.22 (d, *J* = 1.1 Hz, 1H), 5.57 (d, *J* = 9.3 Hz, 1H), 5.42 (d, *J* = 6.0 Hz, 1H), 5.29 (d, *J* = 5.0 Hz, 1H), 5.28 (s, 2H), 5.16 (d, *J* = 5.5 Hz, 1H), 4.63 (t, *J* = 5.7 Hz, 1H), 3.78 (td, *J* = 9.1, 6.1 Hz, 1H), 3.73–3.68 (m, 1H), 3.47–3.44 (m, 2H), 3.40 (td, *J* = 8.9, 5.0 Hz, 1H), 3.24 (td, *J* = 9.1, 5.6 Hz, 1H), 2.40 (d, *J* = 0.9 Hz, 3H). ^13^C NMR (151 MHz, DMSO-*d*_6_) δ 161.66, 160.75, 155.80, 144.77, 141.91, 130.01, 125.14, 113.33, 113.16, 113.07, 102.01, 86.46, 80.84, 73.58, 70.90, 66.69, 62.07, 61.55, 18.67. ESI-MS (*m*/*z*): 442.1 [M + Na] ^+^; HRMS (ESI): Calcd for [M + Na] ^+^ C_19_H_21_N_3_NaO_8_: 442.1262, Found 442.1233.

*7-[(1-β-. D-glucosaminogly-1H-1,2,3-triazol-4-yl) methoxy]-4-methyl-2H-chromen-2-one* (**10l**). m.p. 126.2–128.4 °C; ^1^H NMR (600 MHz, DMSO-*d*_6_) δ 8.50 (s, 1H), 7.71 (d, *J* = 8.8 Hz, 1H), 7.17 (d, *J* = 2.3 Hz, 1H), 7.06 (dd, *J* = 8.8, 2.4 Hz, 1H), 6.23 (s, 1H), 5.51 (d, *J* = 9.5 Hz, 1H), 5.45 (s, 1H), 5.31–5.25 (m, 3H), 4.67 (s, 1H), 3.68 (dd, *J* = 11.4, 3.5 Hz, 1H), 3.46 (dd, *J* = 11.6, 5.3 Hz, 1H), 3.44–3.41 (m, 1H), 3.27 (dt, *J* = 28.2, 8.9 Hz, 2H), 3.13 (t, *J* = 9.3 Hz, 1H), 2.41 (s, 3H), 1.52 (s, 2H). ^13^C NMR (151 MHz, DMSO-*d*_6_) δ 161.52, 160.61, 155.15, 153.90, 142.47, 127.03, 124.78, 113.87, 113.03, 111.78, 102.05, 89.01, 80.55, 77.46, 70.07, 62.08, 61.19, 56.67, 18.62. ESI-MS (*m*/*z*): 441.1 [M + Na] ^+^; HRMS (ESI): Calcd for [M + Na] ^+^ C_19_H_22_N_4_NaO_7_: 441.1386, Found 441.1405.

*7-[(1-β-. D-xylopyranosyl-1H-1,2,3-triazol-4-yl) methoxy]-4-methyl-2H-chromen-2-one* (**10m**). m.p. 122.4–123.9 °C; ^1^H NMR (600 MHz, DMSO-*d*_6_) δ 8.48 (s, 1H), 7.70 (d, *J* = 8.8 Hz, 1H), 7.16 (d, *J* = 2.5 Hz, 1H), 7.05 (dd, *J* = 8.8, 2.5 Hz, 1H), 6.23 (d, *J* = 1.1 Hz, 1H), 5.51 (d, *J* = 9.3 Hz, 1H), 5.43 (d, *J* = 6.1 Hz, 1H), 5.32 (d, *J* = 4.9 Hz, 1H), 5.28 (s, 2H), 5.18 (d, *J* = 5.0 Hz, 1H), 3.83 (dd, *J* = 11.1, 5.3 Hz, 1H), 3.78 (td, *J* = 9.1, 6.1 Hz, 1H), 3.48 (ddd, *J* = 14.2, 10.4, 5.2 Hz, 1H), 3.38 (d, *J* = 10.9 Hz, 1H), 3.35 (t, *J* = 4.4 Hz, 1H), 2.40 (d, *J* = 0.9 Hz, 3H). ^13^C NMR (151 MHz, DMSO-*d*_6_) δ 161.49, 160.60, 155.14, 153.89, 142.41, 127.01, 124.59, 113.87, 113.06, 111.79, 102.05, 89.98, 88.62, 77.52, 73.28, 72.46, 72.30, 69.57, 69.52, 68.82, 64.87, 62.04, 18.61. ESI-MS (*m*/*z*):412.1 [M + Na] ^+^; HRMS (ESI): Calcd for [M + Na] ^+^ C_18_H_19_N_3_NaO_7_: 412.1121, Found 412.1142.

*7-[(1-β-. L-arabinopyranosyl-1H-1,2,3-triazol-4-yl) methoxy]-4-methyl-2H-chromen-2-one* (**10n**). m.p. 129.3–132.5 °C; ^1^H NMR (600 MHz, DMSO-*d*_6_) δ 8.42 (s, 1H), 7.70 (d, *J* = 8.8 Hz, 1H), 7.17 (d, *J* = 2.3 Hz, 1H), 7.06 (dd, *J* = 8.8, 2.4 Hz, 1H), 6.23 (s, 1H), 5.45 (d, *J* = 9.1 Hz, 1H), 5.30 (d, *J* = 3.8 Hz, 3H), 5.04 (d, *J* = 5.8 Hz, 1H), 4.81 (d, *J* = 4.5 Hz, 1H), 4.05 (td, *J* = 9.2, 6.1 Hz, 1H), 3.81 (d, *J* = 1.6 Hz, 1H), 3.77 (d, *J* = 12.0 Hz, 2H), 3.57 (ddd, *J* = 9.2, 5.8, 3.3 Hz, 1H), 2.40 (s, 3H). ^13^C NMR (151 MHz, DMSO-*d*_6_) δ 161.51, 160.61, 155.15, 153.89, 142.45, 127.01, 124.30, 113.86, 113.05, 111.78, 102.06, 88.88, 73.70, 69.81, 69.79, 68.83, 62.03, 18.61. ESI-MS (*m*/*z*):412.1 [M + Na] ^+^; HRMS (ESI): Calcd for [M + Na] ^+^ C_18_H_19_N_3_NaO_7_: 412.1121, Found 412.1139.

*7-[(1-β-. L-rhamnosyl-1H-1,2,3-triazol-4-yl) methoxy]-4-methyl-2H-chromen-2-one* (**10o**). m.p. 125.5–127.7 °C; ^1^H NMR (600 MHz, DMSO-*d*_6_) δ 8.33 (s, 1H), 7.70 (d, *J* = 8.8 Hz, 1H), 7.17 (d, *J* = 2.4 Hz, 1H), 7.06 (dd, *J* = 8.8, 2.4 Hz, 1H), 6.23 (s, 1H), 6.02 (s, 1H), 5.35 (d, *J* = 5.2 Hz, 1H), 5.28 (s, 2H), 5.03 (t, *J* = 5.7 Hz, 2H), 3.90–3.87 (m, 1H), 3.58 (ddd, *J* = 8.9, 5.5, 3.1 Hz, 1H), 3.48 (dq, *J* = 9.3, 6.1 Hz, 1H), 2.40 (s, 3H), 1.22 (d, *J* = 6.1 Hz, 3H). ^13^C NMR (151 MHz, DMSO-*d*_6_) δ 161.55, 160.61, 155.15, 153.89, 142.01, 126.99, 124.98, 113.83, 113.05, 111.76, 102.04, 86.40, 75.45, 73.36, 71.67, 71.02, 61.96, 18.61, 18.31. ESI-MS (*m*/*z*): 426.1 [M + Na] ^+^; HRMS (ESI): Calcd for [M + Na] ^+^ C_19_H_21_N_3_NaO_7_: 426.1277, Found 426.1292.

*7-[(1-β-. D-ribofuranosyl-1H-1,2,3-triazol-4-yl) methoxy]-4-methyl-2H-chromen-2-one* (**10p**). m.p. 123.2–125.6 °C; ^1^H NMR (600 MHz, DMSO-*d*_6_) δ 8.47 (s, 1H), 7.70 (d, *J* = 8.8 Hz, 1H), 7.16 (d, *J* = 2.3 Hz, 1H), 7.05 (dd, *J* = 8.8, 2.3 Hz, 1H), 6.23 (s, 1H), 5.67 (d, *J* = 9.0 Hz, 1H), 5.29 (s, 2H), 5.19–5.15 (m, 2H), 4.92 (d, *J* = 5.3 Hz, 1H), 4.02 (d, *J* = 7.4 Hz, 2H), 3.72 (s, 1H), 3.71–3.68 (m, 1H), 3.61 (dd, *J* = 9.5, 4.4 Hz, 1H), 2.40 (s, 3H). ^13^C NMR (151 MHz, DMSO-*d*_6_) δ 161.49, 160.60, 155.13, 153.88, 142.36, 127.00, 124.75, 113.86, 113.04, 111.78, 102.03, 85.39, 71.54, 69.65, 66.89, 65.65, 62.08, 18.60. ESI-MS (*m*/*z*): 412.1 [M + Na] ^+^; HRMS (ESI): Calcd for [M + Na] ^+^ C_18_H_19_N_3_NaO_7_: 412.1131, Found 412.1223.

### 3.2. Biological Activity

#### 3.2.1. CA Inhibition

The esterase assay was used for testing the inhibitory activities of target compounds against carbonic anhydrases [39]. Firstly, the substrate (4-nitrophenyl acetate) was dissolved in DMSO and the concentration of solution of was 2 mM at 25 °C. Next, the inhibitor was also dissolved in DMSO and the concentration of solution of was 200 mM at 25 °C. Next, the stock solution of inhibitor was diluted to the specified concentration by using analysis buffer. In this case, 10 different inhibitor concentrations have been prepared, varying between 5 mM and 0.5 nM. In addition, the purchased CA was diluted with analysis buffer and added to 96-well plates. Next, different concentrations of inhibitors were also added to 96-well plates. To form the inhibitors and enzymes complex, E-I solutions were preincubated together for 4 h before the test at room temperature. Meanwhile, the nonenzymatic hydrolysis rates were tested by the blank control experiments. The absorbances were determined by using multifunctional enzyme marking instrument.

#### 3.2.2. Cell Viability Assay

The two human cancer cell lines were cultivated for 24 h at 37 °C in a 96-well plate. Subsequently, different concentrations of inhibitors were added to 96-well plate and then was incubated for 96 hours. After the MTT solution was added to per well, cancer cells were further incubated for 4 h at 37 °C. Finally, formazan crystals were dissolved by using DMSO. The cancer cells were from American Type Culture Collection.

#### 3.2.3. Metabolic Stabilities Assay

The metabolic stability of target compounds was tested by predicting intrinsic clearance rate of mouse liver microsomes. The liver microsomes were extracted from mouse. The test compounds and control working solutions were prepared. Meanwhile, the NADPH cofactors were prepared by using NADPH powder. Next, the empty plates were pre-warmed for 10 min minutes and the liver microsomes were diluted with 100 mM phosphate buffer to 0.56 mg/mL. Next, transfer microsome working solutions into pre-warmed incubation plates. In addition, the incubation plates were pre-incubated for 10 min at 40 °C with constant shaking. Then the blank plates were added liver microsomes, NAPDH cofactor and the stop solutions to quench the blank plates. Compound working solution containing microsomes was added to the incubation plates. Furthermore, the NCF60 plates were added buffer and mixed thoroughly 3 times. Then shake plates and incubate for 60 min at 40 °C. The quenching solutions were added to quenching plate T0 and incubation plate (T60) was added NAPDH cofactor, mixing 3 times thoroughly, and immediately remove mixture to quenching plate. Then shake plates and incubate for 60 min at 40 °C. The final each component in incubation medium include liver microsome protein, 1 μΜ test compound, 1 μΜ control compound, 0.99% MeOH and 0.01% DMSO. At 5, 10, 20, 30, and 60 min, quenching solution was added to plates. All sampling plates were shaken and centrifuged. The supernatant was transferred into HPLC water. Each plate was sealed and shaken for 10 min prior to LC-MS/MS analysis.

#### 3.2.4. Plasma Stability Assay

Rat plasma was collected from adult male SD rats. The in vitro stability of **10a** was determined in rat plasma and in PBS at 37 °C, at the meantime, samples were taken at time points 0, 10, 30, 60, and 120, and tested by HPLC. The rest of the time was incubated in a constant temperature shaker to compare different time points. Compare the peak areas of **10a** in PBS or in rat plasma (pH = 7.4, 37 °C) before and after constant temperature incubation at different time points. The compounds in rat plasma were extracted with methanol, centrifuged, and repeated three times, and then detected with a preparative liquid phase at 254 wavelength.

#### 3.2.5. Inhibition Evaluation on hERG K^+^ Channel

HEK 293 cells were stably transfected with human Ether-a-gogo related gene (hERG) channel. The voltage-gated hERG potassium channel current was recorded at room temperature from randomly selected transfected cells using whole-cell recording technique with an EPC10 USB (HEKA) or Multiclamp 700 B amplifier (Molecular Devices) while electrical data was digitalized by Digidata1440 A with acquisition rate of 10 kHz and signals filtered at 2.5 KHz using Patchmaster or pClamp 10, respectively. Dofetilide was included as a positive control to ensure the accuracy and sensitivity of the test system. All experiments were performed in 3 times.

#### 3.2.6. Acute Toxicity Assay

Groups of mice (5 per group) were treated intraperitoneally with target compounds. After the compound was administered, the latency of death was observed and recorded over a span of 14 days. The experimental protocols were evaluated and approved by the ethics committee of the Shenyang Pharmaceutical University.

#### 3.2.7. Experimental Protocol for Docking Study

The molecular docking studies were performed using crystallographic structure of CA IX complex (PDB: 3IAI). The crystallographic protein was obtained by removing all bound water and ligands. In this study, the molecular docking was performed by inserting target compounds into the active pocket of CA IX. The interaction models between protein with target compounds were analyzed after the end of molecular docking.

## 4. Conclusions

In this article, different sugars were connected to coumarins using a 1,2,3-triazole linker for the first time. Thus, 16 new carbohydrate-based coumarin derivatives were synthesized by our optimized one-pot click chemistry reaction. The in vitro CA inhibitory activities of the new derivatives had been tested and the result showed that almost all carbohydrate-based coumarin derivatives showed potent inhibitory activity against hCA IX, with the values of IC_50_ ranging from 11 to 132 nM. Intriguingly, all the compounds showed better CA IX inhibitory activity than initial coumarin fragments. Among all newly synthesized compounds, compound **10a** was found to be the most potent and selective CA IX inhibitor (selectivity index in favor of CA I and CA II equal to 3018, 1955, respectively). Meanwhile, they efficiently reduce tumor cell viability and the extracellular acidification in two different cell lines (HT-29 and MDA-MB-231). In addition, representative compounds were not toxic to normal cell line. In particularly, the target compounds showed good metabolic stability and plasma stability in vitro. Moreover, representative compounds showed low hERG inhibition and low acute toxicity. Furthermore, docking studies were carried out to understand the interactions of our target compounds with the protein target CA IX. The obtained results allowed for the consideration of this compound **10a** as an interesting lead for the development of a new class of selective CA IX inhibitors for the treatment of cancer.

## Data Availability

Not applicable.

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
