# Peer review of "Design, Synthesis and Biological Evaluation of New Carbohydrate-Based Coumarin Derivatives as Selective Carbonic Anhydrase IX Inhibitors via “Click” Reaction"

_molecules, 2022, doi:10.3390/molecules27175464_

Round 1

Reviewer 1 Report

This manuscript describes carbohydrate-based coumarin carbonic anhydrase IX inhibitors by using 1,2,3-triazoles as linker synthesized via one-pot click azide-alkyne cycloaddition. These scaffolds were further assayed for the inhibition of three carbonic anhydrase isoforms (CA I, CA II and 17 CA IX) and display potent inhibitory activity against hCA IX with the values of IC 50 ranging from 11-132 nM. As well as authors carried out docking for understand the interactions of target compounds with the protein target CA IX and they claiming one of those (6a) as an interesting lead for the development of a new class of selective CA IX inhibitors for the treatment of cancer. These studies are also well established by others and thoroughly investigated with different types of cores and novelty not extremely high. In my opinion, this manuscript is suitable for this journal. There are some issues of this manuscript should be clarified before its acceptance.

Comments

1.     The term novel should be new in all cases and in page 2, Coumarins and their isosteres should be mentioned as figure.

2.     Authors should explain this report with previous studies as a figure in detail explanation of their results.

3.     In detail explanation of inhibitory potency of 6a/6c with respect to others vice versa should be provided.

4.     The compound numbering also verified, and they started with the compound 6a first instead of 1 in page 3 and this should be fixed.

5.     What is VcNa ?? line 109/117/122-- t-BuOH should be uniform, in page 13 exp.section authors should mentioned compound nature whether crystalline or white solid?? Yield as well as Rf /solvent systems.

6.     In SI information, authors should mention solvent info/nature of nuclei, solvent peaks and NMR window should be always 0-10 ppm for 1H, 0-210 ppm for 13C, the peaks heights particularly in carbon case should be increased. The nmr values should be rechecked again and I didn’t see peak values properly.

Author Response

Thanks for your comments on our paper. We have revised our paper according to your comments:

This manuscript describes carbohydrate-based coumarin carbonic anhydrase IX inhibitors by using 1,2,3-triazoles as linker synthesized via one-pot click azide-alkyne cycloaddition. These scaffolds were further assayed for the inhibition of three carbonic anhydrase isoforms (CA I, CA II and 17 CA IX) and display potent inhibitory activity against hCA IX with the values of IC50 ranging from 11-132 nM. As well as authors carried out docking for understand the interactions of target compounds with the protein target CA IX and they claiming one of those (6a) as an interesting lead for the development of a new class of selective CA IX inhibitors for the treatment of cancer. These studies are also well established by others and thoroughly investigated with different types of cores and novelty not extremely high. In my opinion, this manuscript is suitable for this journal. There are some issues of this manuscript should be clarified before its acceptance.

  1. The term novel should be new in all cases and in page 2, Coumarins and their isosteres should be mentioned as figure.

According to the suggestion of reviewer, we have changed the word “novel” to “new” in our revised manuscript. Then, we added isosters of the coumarin such as thiocoumarin, 2-thioxocoumarin, and dithiocoumarin to our revised manuscript. Meanwhile, the inhibitory activity of isosters, relevant description and references had also been added to revised manuscript. Thank you for your suggestion to make our introduction part more complete.

  1. In detail explanation of inhibitory potency of 6a/6c with respect to others vice versa should be provided.

Thanks for the reviewer’s kind suggestion. As we all know, at present, several coumarin structures have been reported to have carbonic anhydrase inhibitory activity. Among all the reported coumarins, 7-hydroxy coumarin and 4-methyl-7-hydroxy coumarin are not only readily available, but also serves as the starting fragment of our target compounds. Therefore, 7-hydroxy coumarin and 4-methyl-7-hydroxy coumarin were chosen as representative compounds to compare the activity of our compounds with those reported by other research groups. In our study, the tumor-associated isoform hCA IX was weakly inhibited by parent coumarins umbelliferone and 4-methylumbelliferone. Notably, the introduction of sugar moieties into the coumarin ring at position 7 lead to a significant increase of the hCA IX inhibitory potency with IC50 in the range of 0.011-0.132 μM. By using umbelliferone or 4-methylumbelliferone as model compounds, we could compare easily the activity of our compounds with others. In detail explanation of inhibitory potency of 6a/6c with respect to umbelliferone or 4-methylumbelliferone had been added to our revised manuscript. Thanks again.

  1. The compound numbering also verified, and they started with the compound 6a first instead of 1 in page 3 and this should be fixed.

Thanks for the reviewer’s kind suggestion. We renumbered the compounds in the order in which they appeared. Meanwhile, the number of the compounds in the supporting information had also been changed. Thanks!

  1. What is VcNa ?? line 109/117/122-- t-BuOH should be uniform, in page 13 exp. section authors should mentioned compound nature whether crystalline or white solid?? Yield as well as Rf /solvent systems.

Thanks for the reviewer’s kind suggestion. VcNa is sodium ascorbate. t-BuOH had been uniform. In addition, compound nature, yield as well as solvent systems had been added to our revised manuscript.

  1. In SI information, authors should mention solvent info/nature of nuclei, solvent peaks and NMR window should be always 0-10 ppm for 1H, 0-210 ppm for 13C, the peaks heights particularly in carbon case should be increased. The nmr values should be rechecked again and I didn’t see peak values properly.

Thanks for the reviewer’s kind suggestion. According to the suggestion of reviewer, we had revised SI information. The NMR windows had been revised. The peaks heights in carbon case had also been increased. To make the peak values visible, we resized the font and image size. Thanks again.

Reviewer 2 Report

This research paper reported the discovery of sugar-containing coumarins as selective carbonic anhydrase inhibitors. The design (via a click reaction) as well as the activities of these compounds is highlighted. The manuscript is well-organized, while some key experiments should be supplemented to support the results and conclusions in this study.

Major points:

1. The previously reported sugar-containing coumarins as carbonic anhydrase inhibitors should be introduced, and their structures as well as activities should be presented.

2. The binding affinity of the most potent compound 6a to hCA IX is required to be tested to verify the enzymatic activity.

3. The cellular toxicity of the most potent compound 6a is required to be evaluated in normal mammalian cells.

4. The very strong enzymatic activity (nM level), while the weak cellular antitumor activities (high μM) should be discussed.

5. The negative control (compound with weak enzymatic activity in this study) should be included in the extracellular pH measurement assay.

6. The positive control should be included in the hERG assay to validate the experiment conditions.

Minor points:

1. The interactions between 6a and the protein in the docking study are suggested to be described with figure illustration. Especially, the binding comparison between the current molecules (e.g., 6a) and the previous reported sugar-containing coumarins should be included to highlight the significance of the current study.

2. In the supplementary, the whole H-/C-NMR spectrum (showing the chemical shift starting from 0) are required. Some of the current H-NMR spectrum started from 2 ppm, and C-NMR spectrum started from 15 or 20 ppm. Please check all and revise them.

Author Response

Thanks for your comments on our paper. We have revised our paper according to your comments:

This research paper reported the discovery of sugar-containing coumarins as selective carbonic anhydrase inhibitors. The design (via a click reaction) as well as the activities of these compounds is highlighted. The manuscript is well-organized, while some key experiments should be supplemented to support the results and conclusions in this study.

  1. The previously reported sugar-containing coumarins as carbonic anhydrase inhibitors should be introduced, and their structures as well as activities should be presented.

Thanks for the reviewer’s kind suggestion. The previously reported sugar-containing coumarins had been added to our revised manuscript. Meanwhile, the inhibitory activity of isosters, relevant description and references had also been added to revised manuscript.

  1. The binding affinity of the most potent compound 6a to hCA IX is required to be tested to verify the enzymatic activity.

According to the suggestion of reviewer, the CA IX inhibitory activities of the representative coumarins containing triazoles were measured by a stopped-flow, CO2 hydration assay method. The results showed the target compounds possessed the same order of magnitude of inhibitory activity under both test conditions.

  1. The cellular toxicity of the most potent compound 6a is required to be evaluated in normal mammalian cells.

According to the suggestion of reviewer, the cellular toxicity of the most potent compounds 6a and 6c against normal mammalian cell had been tested. The data suggested the carbohydrate-based coumarins had no apparent cytotoxicity toward MCF-10A cell line (IC50 >100 μM). Thanks for the reviewer’s suggestion.

  1. The very strong enzymatic activity (nM level), while the weak cellular antitumor activities (high μM) should be discussed.

As the reviewer said, our target compounds showed strong enzymatic activity, but weak cellular activity. This result was also found in our previous study. This result is closely related to the function of carbonic anhydrase IX. And as we all know, CA IX promotes the acidification of the extracellular matrix, which is conducive to the growth and metastasis of tumor cells. Therefore, these hCA IX inhibitors are not potent enough in anti-tumor monotherapy only depending on the regulation of tumor microenvironment. Therefore, more pharmacological effects of CA IX inhibitors as antitumor agents need to be further explored. According to the suggestion of reviewer, we had added the discussion in our revised manuscript.

  1. The negative control (compound with weak enzymatic activity in this study) should be included in the extracellular pH measurement assay.

Based on the suggestion of reviewer, we had added the negative control (compound 10h with weak enzymatic activity in this study) to our revised manuscript. Thanks!

  1. The positive control should be included in the hERG assay to validate the experiment conditions.

According to the suggestion of reviewer, the positive control experiment had been added in the experimental. Thanks again!

Minor points:

  1. The interactions between 6a and the protein in the docking study are suggested to be described with figure illustration. Especially, the binding comparison between the current molecules (e.g., 6a) and the previous reported sugar-containing coumarins should be included to highlight the significance of the current study.

According to the suggestion of reviewer, we complemented the molecular docking experiments of previous reported sugar-containing coumarins 8 and 9. Then, we compared the binding modes between our most potent compounds and the previous reported sugar-containing coumarins. The results showed the sugar groups of our compounds could penetrate deep into the hydrophilic part to form sufficient binding after introducing the 1,2,3-trizaole group. Thanks for the reviewer’s comments, which further highlight the importance of significance of our study.

  1. In the supplementary, the whole H-/C-NMR spectrum (showing the chemical shift starting from 0) are required. Some of the current H-NMR spectrum started from 2 ppm, and C-NMR spectrum started from 15 or 20 ppm. Please check all and revise them.

Thanks for the reviewer’s kind suggestion. According to the suggestion of reviewer, the NMR windows had been corrected. 

Round 2

Reviewer 1 Report

The authors addressed most of the concerns in the revised version and i have no more comments and happy to recommend it for acceptance.

with best wishes

Reviewer 2 Report

After revision, the quality of this manuscript was significantly improved, and can reach the required quality standard of Molecules in my opinion. I recommend accepting without further revisions.